# List-aware Reranking-Truncation Joint Model for Search and Retrieval-augmented Generation

## ABSTRACT

The results of information retrieval (IR) are usually presented in the form of a ranked list of candidate documents, such as web search for humans and retrieval-augmented paradigm for large language models (LLMs). List-aware retrieval aims to capture the list-level contextual features to return a better list, mainly including reranking and truncation. Reranking finely re-scores the documents in the list. Truncation dynamically determines the cut-off point of the ranked list to achieve the trade-off between overall relevance and avoiding misinformation from irrelevant documents. Previous studies treat them as two separate tasks and model them separately. However, the separation is not optimal. First, it is hard to share information between the two tasks. Specifically, reranking can provide fine-grained relevance information for truncation, while truncation can provide utility requirement for reranking. Second, the separate pipeline usually meets the error accumulation problem, where the small error from the reranking stage can largely affect the truncation stage. To solve these problems, we propose a Reranking-Truncation joint model (GenRT) that can perform the two tasks concurrently. GenRT integrates reranking and truncation via generative paradigm based on encoder-decoder architecture. We also design the novel loss functions for joint optimization to make the model learn both tasks. Sharing parameters by the joint model is conducive to making full use of the common modeling information of the two tasks. Besides, the two tasks are performed concurrently and co-optimized to solve the error accumulation problem between separate stages. Experiments on public learning-to-rank benchmarks and open-domain Q&A tasks show that our method achieves SOTA performance on both reranking and truncation tasks for web search and retrieval-augmented LLMs. To the best of our knowledge, this is the first work that discusses list-aware retrieval (esp. truncation task) in retrieval-augmented LLMs.

## CCS CONCEPTS

• **Information systems → Retrieval models and ranking**.

## KEYWORDS

Reranking, Truncation, Retrieval-augmented LLMs

**ACM Reference Format:**
Anonymous Author(s). 2024. List-aware Reranking-Truncation Joint Model for Search and Retrieval-augmented Generation. In *Proceedings of the ACM Web Conference 2024 (www '24), May 13–17, 2024, Singapore.* ACM, New York, NY, USA, 10 pages. https://doi.org/10.1145/nnnnnnn.nnnnnnn

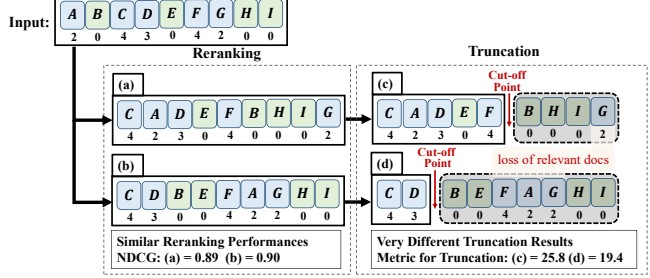

**Figure 1: Problems of the separation of reranking and truncation. Separate pipeline leads to the error accumulation problem between two stages and the loss of relevant documents. Similar reranking but results in different truncations.**

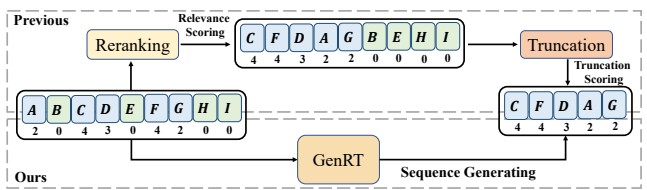

**Figure 2: Comparison with previous methods.**

## 1 INTRODUCTION

In information retrieval (IR), even though the ranking methods based on probability ranking principle [39] (PRP) that assumes the relevance of each document is modeled independently for a query have been widely used [9, 22, 40], many studies have shown that users' feedback of the retrieval result is based on the entire returned list [41, 47, 48]. In web search, humans usually compare multiple documents in the list before clicking. In retrieval-augmented paradigm for LLMs, LLMs process the documents in the list via self-attention [42] and select the information for generation [23, 27, 46]. It has been proven that the performance of the retrieval-augmented LLMs is affected by the length of the retrieved list and the arrangement of documents within the list provided in the prompt [27].

Therefore, list-aware retrieval models [31] are proposed as the post-processing stage of IR, which are used to capture the list-level contextual features. List-aware retrieval mainly includes reranking and truncation. Reranking exploits list-level contextual features to re-score each document. Truncation dynamically determines the cut-off point of the list to achieve the optimal trade-off between overall relevance and weeding out irrelevant documents, which is meaningful to improve retrieval efficiency and avoid misinformation [27, 43]. Truncation is important for the domains that need users to use the high cost to judge the relevance of documents [29]. It is also important for retrieval-augmented LLMs. Because the

performance of LLMs fluctuates with the number of retrieved documents, while for different prompts, the suitable number of retrieved documents changes dynamically. Blindly increasing the number of retrieved documents will not always improve performance, but will affect the efficiency of LLMs and introduce noise [27].

Previous methods model reranking and truncation separately [1, 4, 31, 43], first rerank and then truncate. Although LeCut [29] exchanges features between the two models in training, it still treats them as separate models and stages during inference. This leads to several problems. **First**, these two tasks are interdependent but the separation makes it hard to exploit the information shared between them. Document relevance modeled by reranking can provide an important basis for truncation. Trade-off characterization of the relevance and position of documents in the list modeled by truncation provides important contextual interaction information for reranking. **Second**, the separate pipeline usually meets the error accumulation problem, where the error from the reranking stage can affect the truncation stage largely, which cannot be directly optimized during training. There are two factors that cause this. **1)** Inconsistent document relevance judgment in two separate stages. As shown in list (b) of Figure 1, the reranking model mistakenly thinks that $B$ and $E$ are highly relevant, but the truncation model thinks they are irrelevant and thus truncates the list at top-ranked position. **2)** Reranking and truncation have different concerns to ranking list. Truncation is more sensitive to the ranking performance of the top-ranked documents because once too many irrelevant documents appear at the top of list, it causes the list to be truncated at these documents to lose relevant documents that are ranked behind it. But reranking focuses on the overall ranking performance of the entire list and is not sensitive to the case that irrelevant documents appear at the top of list. Figure 1 shows that although (a) and (b) have similar reranking performance (0.89 and 0.90), two irrelevant top-ranked documents ($B$ and $E$) of (b) result in the worse result of reranking-truncation pipeline than (a) (19.4 < 25.8).

To solve the above problems, it is necessary to get a joint model to perform them concurrently. However, there are several challenges. First, how to make the two tasks share the modeling information effectively (**C1**). Second, reranking is a process of dynamically changing the ranking list. However, the truncation decision needs to be based on a static list, how to perform them concurrently (**C2**). Third, how to design loss functions for joint learning (**C3**).

In this paper, we propose a Reranking-Truncation joint model via sequence generation called GenRT (shown in Figure 2). The input of GenRT is a ranked list, and GenRT can perform reranking and truncation concurrently to directly output the final list that has been reranked and truncated. Specifically, to address **C1**, we design the global dependency encoder to provide global list-level contextual features within ranked lists that can be shared by reranking and truncation. To address **C2**, different from the mainstream ranking model that ranks documents by estimating the score of documents, GenRT outputs the final reranked list step by step in the paradigm of sequence generation. At each time step, the document at the current ranking position is selected according to the previous state and the current candidate set, and the local optimal truncation decision is made at the same time. Truncation is transformed into a binary classification task based on the forward and backward sequential information of the dynamic list at each step. Sequence

generation paradigm records the forward information and we also introduce the local backward window to provide the backward information. In this way, our model can combine dynamic reranking with static truncation. To address **C3**, we design step-adaptive attention loss and step-by-step lambda loss and combine them as the objective function for reranking. We introduce the reward augmented maximum likelihood (RAML [30]) to design the RAML-based soft criterion as the loss function for truncation at each step.

To sum up, our contributions are: (1) We point out the problem of separating reranking and truncation in list-aware retrieval and propose that these two tasks can be concurrently done with a joint model. (2) We propose the novel model, inference paradigm, and loss function to jointly optimize and perform reranking and truncation on only one model. (3) Experimental results on public learning-to-rank benchmarks and open-domain Question-answer tasks show that our method achieves state-of-the-art performance on both reranking and truncation tasks for web search and retrieval-augmented LLMs. The code will be released on GitHub.

## 2 RELATED WORK

### 2.1 Reranking in List-aware Retrieval

Reranking in list-aware retrieval exploits list-level contextual features to re-score and rank each document in the list. DLCM [1] uses recurrent neural network (RNN) to encode the contextual information of the list and reranks the documents. GSF [2] proposes a multivariate scoring function framework to score the document affected by other documents. SetRank [31] employs multi-head self-attention to capture interaction information within the list. PRM [32] optimizes personalized recommendations by capturing user-personalized information. IRGPR [28] employs GNN to capture the relationship between candidate items. DASALC [35] further explores neural IR models from data augmentation perspective. SRGA [33] proposes a scope-aware reranking model with gated attention. MIR [45] considers the dynamic interaction between the user behavior and the candidate set. [13, 14, 17, 44] exploit counterfactual signals for re-score. Different from them, we use the sequence generation method to directly generate a reranked list. Although Seq2Slate [5] and Globalrerank [49] also use the similar generative method, they do not satisfy the permutation-invariant [31]. The most prominent difference between previous studies is that our method can not only be applied to single reranking task but also perform reranking and truncation concurrently.

### 2.2 Truncation in List-aware Retrieval

Truncation aims to determine the best cut-off point for the input ranked list to achieve the optimal trade-off between overall relevance and weeding out irrelevant documents. Recently, some work uses machine learning methods to solve the truncation problem. [11] investigate machine learning approaches for learning dynamic cut-offs within cascade-style IR systems. BiCut [26] leverages bidirectional LSTM to find the best truncation point. Choppy [4] uses transformer to model the input ranked list. AttnCut [43] uses RAML [30] to make the model optimize user-defined metric directly and smoothly. LeCut [29] passes the relevance information of the ranking model to the truncation model during training time and iteratively trains between ranking and truncation. Different

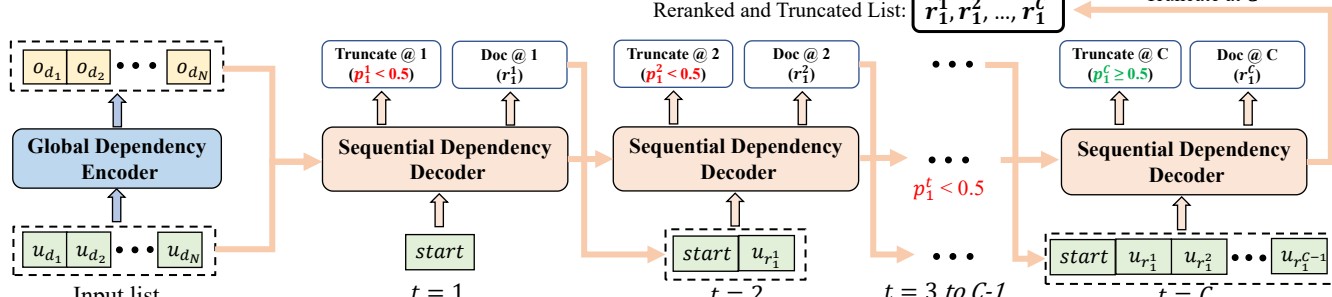

**Figure 3: Overview of GenRT. Global dependency encoder captures the features of the initial list. Sequential dependency decoder generates the final list step-by-step with decreasing relevance and concurrently makes truncation decision.**

from them, we focus on jointly modeling truncation and reranking. We transform truncation into a step-by-step binary classification task and leverage the paradigm of sequence generation to combine dynamic reranking with static truncation and design binary classification soft criterion as the optimization object for truncation.

## 3 METHOD

The overall architecture of GenRT is shown in Figure 3. GenRT aims to jointly model reranking and truncation by a shared model and perform these two tasks concurrently during the inference. The most critical challenge to achieve this is that reranking dynamically changes the ranked list, while the truncation decision needs to be based on a static list. To address the challenge, GenRT adopts encoder-decoder architecture consisting of a global dependency encoder and a sequential dependency decoder. Global dependency encoder is used to capture the global list-level contextual features of the input list by multi-head self-attention (MHSA) [42], which can be shared by the reranking and truncation. Sequential dependency decoder generates the final list step-by-step with decreasing relevance and makes truncation decision at each step based on the bidirectional sequential information. This sequence generation paradigm combines dynamic reranking with static truncation, which can address the critical challenge. Details are introduced below.

### 3.1 Global Dependency Encoder

Global dependency encoder captures the list-level contextual features within the input list. As for the input of the encoder, each document in the input list is represented as an embedding. As shown in Figure 4, given a query $q$, a document list $D = [d_1, d_2, ..., d_N]$, and initial ranking score list $L = [l_1, l_2, ..., l_N]$ obtained from previous ranking stage (e.g. retrieval), input embedding for document $d_i$ is:

$$\mathbf{u}_{d_i} = f(q, d_i, l_i), \mathbf{u}_{d_i} \in \mathbb{R}^Z, i \in [1, N]. \tag{1}$$

$N$ is the number of documents in the input list, $f$ is used to fuse the features of $q$, $d_i$ and the initial ranking feature $l_i$. Specifically, for the feature-based ranking tasks such as MSLR (input data is the learning-to-rank feature), we follow [1, 31] to use the traditional learning-to-rank method to extract the features (matching, pagerank, etc.) between $q$ and $d_i$ as described in SetRank [31] and concatenate the features with the ranking score $l_i$ to obtain $\mathbf{u}_{d_i}$. For the text-based ranking tasks such as Natural Questions [25] (input

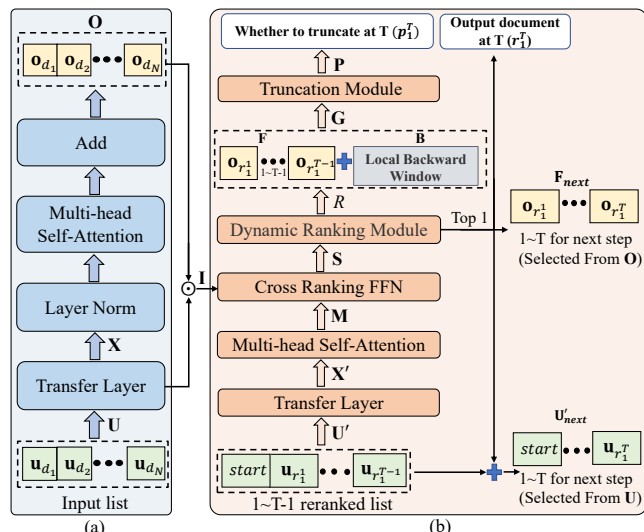

**Figure 4: (a) Global Dependency Encoder and (b) Sequential Dependency Decoder at T-th step.**

data is text), we use the output embedding of [CLS] token in the interaction-based ranking model as the representation ($C(q, d_i)$) for $q$ and $d_i$. The score $l_i$ is mapped to a learnable position embedding $\mathbf{lp}_i \in \mathbb{R}^Z$ according to its rank in $L$. Then, $C(q, d_i)$ and $\mathbf{lp}_i$ are added element-wise to obtain $\mathbf{u}_{d_i}$. The embeddings corresponding to the documents in the list are concatenated to get $\mathbf{U} \in \mathbb{R}^{N \times Z}$, and the matrix of vectors for the input list $L$ can be represented as:

$$\mathbf{U} = [\mathbf{u}_{d_1}, \mathbf{u}_{d_2}, ..., \mathbf{u}_{d_N}]^T. \tag{2}$$

Transfer layer is used to map $\mathbf{U}$ to the vector space of multi-head self-attention (MHSA) [42] and align the dimensions:

$$\mathbf{X} = Swish(\text{MLP}(\mathbf{U})), \mathbf{X} \in \mathbb{R}^{N \times E}, \tag{3}$$

where MLP is multilayer perceptron, $Swish$ is the activation function that is shown to have stronger generalization [38], $E$ is the dimension of MHSA. In order to retain the feature of the document itself while capturing the list-level contextual features, we adopt

the residual connection method as:

$$\mathbf{O} = \mathbf{X} + \text{MHSA}(\text{LN}(\mathbf{X})), \tag{4}$$

where $\mathbf{O} = [\mathbf{o}_{d_1}, \mathbf{o}_{d_2}, ..., \mathbf{o}_{d_N}]^T$ is the output of global dependency encoder. $\mathbf{o}_{d_i}$ contains the list-level contextual features in the ranked list and the input feature of $d_i$, which can be shared by reranking and truncation. LN is layer normalization [3].

## 3.2 Sequential Dependency Decoder

Sequential dependency decoder follows the paradigm of generating sequences to generate the final list step-by-step with decreasing relevance and makes truncation decision at each step based on the bidirectional sequential information. In Figure 4, we show the operation of the decoder at the $T$-$th$ step. Multi-head self-attention captures the interaction information of the document sequence from 1 to $T - 1$ steps of the reranked list. Previously generated documents from 1 to $T - 1$ steps serve as sequential dependency information to facilitate the selection of subsequent relevant documents [10, 15]. Cross ranking FFN and dynamic ranking module determine the dynamic ranking list and select the best output document at each step. The ordinal number of the step is the rank of its output document in the final reranked list. At the same time, the truncation module makes truncation decision based on the bidirectional sequential information obtained by the dynamic ranking module and local backward window at each step.

We describe the operation mechanism of the decoder at the $T$-$th$ step in detail. Given an input document list $D' = \{r_1^1, r_1^2, ..., r_1^{T-1}\}$ ($r_1^t$ is the output document at step $t$) from 1 to $T - 1$ (the documents that have been reranked), the representation vectors $U'$ and $X'$ can be obtained according to Equ.(1)(2)(3) with the same parameters. MHSA is used to capture the list-level contextual features of the reranked list from 1 to $T - 1$ and it can be used as the sequential dependency for the current step:

$$\text{MHSA}(\mathbf{X}') \rightarrow [\mathbf{m}_{r_1^1}, \mathbf{m}_{r_1^2}, ..., \mathbf{m}_{r_1^{T-1}}]^T.$$

Cross ranking FFN estimates the generation score for each document to select the best output document at the current generating step, which is the core module of the decoder. The input of this module comes from decoder and encoder. Specifically, the input from the decoder side is $\mathbf{m}_{r_1^{T-1}}$ from MHSA($\mathbf{X}'$), which is the sequential dependency information at $T$-$th$ step. Expand $\mathbf{m}_{r_1^{T-1}} \in \mathbb{R}^E$ to the matrix $\mathbf{M} \in \mathbb{R}^{N \times E}$, each row of $\mathbf{M}$ is $\mathbf{m}_{r_1^{T-1}}$. The input from the encoder side is $\mathbf{I} \in \mathbb{R}^{N \times E}$ that can be obtained by latent cross [6, 35]:

$$\mathbf{I} = (1 + \text{MLP}(\mathbf{O})) \odot \text{FFN-Swish}(\mathbf{U}), \tag{5}$$

where $\mathbf{U}$ and $\mathbf{O}$ are obtained from Equ. (2) and (4) respectively, MLP is multilayer perceptron, FFN-Swish is the block of MLP and Swish, $\odot$ is the element-wise multiplication operator. $\mathbf{I}$ is the embedding matrix of the candidate document set at the current step. $\mathbf{M}$ is the sequential dependency matrix of the current step and is used to interact with the embedding of each document in $\mathbf{I}$ to get the score. Specifically, $\mathbf{I}$ and $\mathbf{M}$ are concatenated and processed by a row-wise FFN (rFFN) to get the predicted score of each document in the candidate set at the current step:

$$\mathbf{S} = \text{rFFN}(\text{Concat}(\mathbf{I}, \mathbf{M})), \mathbf{S} \in \mathbb{R}^N. \tag{6}$$

Dynamic ranking module masks the documents in steps 1 to $T - 1$ (avoid selecting duplicate documents) and ranks the remaining candidate documents according to $\mathbf{S}$ in descending order to get the ranking list $R = \{r_1^T, r_2^T, ..., r_{N-T+1}^T\}$ at the current step. The generated document at current step is the Top-1 element in this list (i.e., $r_1^T$). Document generation is finished and the next is truncation.

Previous truncation models need to be performed on a static list. However, reranking is a process of dynamically changing the ranking list, which is the challenge for the joint model to perform reranking and truncation concurrently (**C2** in Section 1). To address this challenge, we transform truncation into a binary classification task at each step. Truncation module aggregates forward and backward information of the current generated document ($r_1^T$) to make truncation decision. Specifically, the module records the embedding sequence of the selected documents ($D'$) in $\mathbf{O}$ (Equ. 4) from 1 to T-1 as the forward information:

$$\mathbf{F} = [\mathbf{o}_{r_1^1}, \mathbf{o}_{r_1^2}, ..., \mathbf{o}_{r_1^{T-1}}]^T,$$

which is the sequence of the documents that precede the document output at the current step ($r_1^T$) in the reranked list. Reranking-Truncation joint model has to complete the truncation decision when reranking. However, the reranked list is generated step by step with decreasing relevance, when outputting the current document $r_1^T$, the model cannot capture the backward information of the documents ranked behind $r_1^T$ in the final reranked list. To address it, local backward window is proposed to select $\beta$ documents behind $r_1^T$ in the ranking list $R$ at the current step and gets the corresponding embedding sequence from $\mathbf{O}$:

$$\mathbf{B} = [\mathbf{o}_{r_2^T}, \mathbf{o}_{r_3^T}, ..., \mathbf{o}_{r_{\beta+1}^T}]^T.$$

The reason why only $\beta$ documents are selected is that $R$ is only a local ranking list of the current step and cannot represent the global result of reranking, selecting all the remaining documents will introduce noise. Embedding sequences $\mathbf{F}$, $\mathbf{o}_{r_1^T}$, and $\mathbf{B}$ are concatenated to $\mathbf{G} = \text{Concat}(\mathbf{F}, \mathbf{o}_{r_1^T}, \mathbf{B})$ as the input of truncation module.

To distinguish between forward and backward information and the position of document embedding in $\mathbf{G}$, we introduce relative position encoding into MHSA like T5 [36] for truncation module. Specifically, attention calculation with relative position encoding for the $a$-$th$ and $b$-$th$ vectors of the input is $\mathbf{H}_a \mathbf{W}^Q (\mathbf{H}_b \mathbf{W}^K)^T + pos_{a,b}$, where $\mathbf{W}^Q, \mathbf{W}^K$ are matrices in MHSA, $\mathbf{H}_a$ and $\mathbf{H}_b$ are embeddings of input, $pos_{a,b} = bucket(a - b)$, $bucket$ is a bucketing function. We call this module MHSA$_{pos}$, which is used to aggregate the bidirectional sequential information at the current step and make the truncation decision at $r_1^T$. The result of the truncation decision for step $T$ ($r_1^T$) can be obtained by:

$$\text{MHSA}_{pos}(\mathbf{G}) \rightarrow [\mathbf{j}_{r_1^1}, ..., \mathbf{j}_{r_1^{T-1}}, \mathbf{j}_{r_1^T}, \mathbf{j}_{r_2^T}, ..., \mathbf{j}_{r_{\beta+1}^T}]^T,$$
$$\mathbf{P} = \text{Softmax}(\text{MLP}(\mathbf{j}_{r_1^T})), \mathbf{P} \in \mathbb{R}^2, \tag{7}$$

$\mathbf{P} = [p_0, p_1]$ is a binary probability distribution representing the probability of truncating or not at the current step (i.e., at $r_1^T$). If the decision is truncating, GenRT directly returns the documents generated at steps 1 to $T$ as the final reranked and truncated list, if not, the model continues to execute until the decision is truncating or the reranking of all documents is completed.

The document generation and truncation decision at the current step are finished and the sequential dependency state can be passed to the next step by $\mathbf{U}'_{next} = Concat(\mathbf{U}', \mathbf{u}_{r_1^T})$ and $\mathbf{F}_{next} = Concat(\mathbf{F}, \mathbf{o}_{r_1^T})$. For the first step without history dependency, a trainable vector called $start$ is used as the initial input.

The final reranked and truncated list $Res = \{r_1^1, r_1^2..., r_1^\epsilon\}$ is the document sequence generated from step 1 to $\epsilon$, where $\epsilon$ is the first truncated step or the number of documents in the input list, $r_1^t$ indicates the output document at the $t$-$th$ step.

## 3.3 Training and Inference

We design different loss functions for reranking and truncation respectively. For reranking, we design the step-adaptive attention loss (improved from [1]) and step-by-step lambda loss under the generative ranking paradigm and combine them as the optimization objective. Specifically, for $T$-$th$ step, given a query $q$, a candidate document list $D = [d_1, d_2, ..., d_N]$, the relevance label set $Y = \{y_1, y_2, ..., y_N\}$ and set $D'$ in which the documents has been selected, the ground-truth attention score $a_i$ for $d_i$ is assigned as:

$$a_i = \frac{\exp(\phi(d_i))}{\sum_{d_j \in D} \exp(\phi(d_j))}, \phi(d_i) = \begin{cases} -10^4, & d_i \in D'; \\ y_i, & otherwise. \end{cases} \quad (8)$$

For the document scoring matrix of candidate document set $\mathbf{S} = [s_1, s_2, ..., s_N]^T$ (obtained by Equ. (6)) predicted by the model at step $T$, the same attention distribution strategy as Equ.(8) is used to get the predicted attention score $p_i$ for $d_i$. The step-adaptive attention loss at $T$-$th$ step is the cross entropy of the attention distribution:

$$L_{sa\text{-}att}^T = -\sum_{d_i \in D} a_i \log(p_i), \mathcal{L}_{sa\text{-}att} = \sum_{t=1}^N \alpha_t L_{sa\text{-}att}^t, \quad (9)$$

where $t$ is the ordinal number of the step and indicates the rank of the document in reranked list, $\alpha_t = \frac{1}{\log(1+t)}$ makes the model give more optimization weight to the top-ranked documents, $N$ is the number of documents in the input list, i.e. the number of steps performed by the decoder. In addition to $\mathcal{L}_{sa\text{-}att}$, we also design the step-by-step lambda loss (sbs loss). Given an already generated reranking sequence including $\epsilon$ documents[1] $Res' = \{r_1^1, r_1^2, ..., r_1^\epsilon\}$, its relevance label list $Y' = \{y^1, y^2, ..., y^\epsilon\}$ and a list of the scoring matrix for the candidate document set at each step $\mathbb{S} = \{\mathbf{S}^1, \mathbf{S}^2, ..., \mathbf{S}^\epsilon\}$ ($r_1^t, y^t, \mathbf{S}^t$ mean the output document, label of the document and scoring matrix (Equ. (6)) at step $t$ respectively), the sbs loss is described as Algorithm 1. Specifically, sbs loss modifies the scoring matrix of the model at the corresponding step by adding penalty terms to the document pairs with non-decreasing relevance in the generated sequence (for example, in $\mathbf{S}^{t_f}$ at step $t_f$, because $y^{t_b}$ is bigger than $y^{t_f}$, $s_{t_b}$ should be bigger than $s_{t_f}$, so that $r_1^{t_b}$ can be ranked before $r_1^{t_f}$), so as to give priority to generating high-relevance documents. The loss function for $\mathcal{L}_{sbs}$ is:

$$\mathcal{L}_{sbs} = \sum_{i=1}^\epsilon \sum_{j=i+1}^\epsilon \mathbb{I}(y^j > y^i) \Delta N \log(1 + e^{s_i - s_j}),$$

where $s_j(s_i)$ is the predicted score at step $i$ for $r_1^j(r_1^i)$ in $\mathbf{S}^i$, $\Delta N$ is the Lambda Loss ($NDCG_{swap} - NDCG$). The loss function for

---

**Algorithm 1** Step-by-step lambda loss.

**Input**: $Res, Y', \mathbb{S}, \epsilon$; **Output**: $\mathcal{L}_{sbs}$
1: Let $\mathcal{L}_{sbs} = 0$.
2: **for** $t_f = 1$ to $\epsilon$ **do**
3:     **for** $t_b = t_f + 1$ to $\epsilon$ **do**
4:         **if** $y^{t_b} > y^{t_f}$ **then**
5:             $r_1^{t_b}$ is ranked behind $r_1^{t_f}$ but more relevant than $r_1^{t_f}$.
6:             $\Delta NDCG = NDCG_{swap} - NDCG$   (*Lambda Loss*)
7:             $s_{t_f}$ is the predicted score at step $t_f$ for $r_1^{t_f}$ from $S^{t_f}$
8:             $s_{t_b}$ is the predicted score at step $t_f$ for $r_1^{t_b}$ from $S^{t_f}$
9:             $\mathcal{L}_{pair} = \log(1 + \exp(s_{t_f} - s_{t_b}))$.
10:            $\mathcal{L}_{sbs} += \Delta NDCG \times \mathcal{L}_{pair}$
11:         **end if**
12:     **end for**
13: **end for**
14: **return** $\mathcal{L}_{sbs}$

---

reranking is the combination of $\mathcal{L}_{sa\text{-}att}$ and $\mathcal{L}_{sbs}$:

$$\mathcal{L}_R = \mathcal{L}_{sa\text{-}att} + \eta \mathcal{L}_{sbs},$$

where $\eta \in [0, 1]$ is the hyperparameter used to tune the weights.

For truncation, which makes binary decision based on bidirectional sequential information at each step, we define the binary soft label for each step based on RAML [30] and compute loss by maximum likelihood estimation (MLE) criterion. Specifically, we introduce the metric for truncation [20] and call it TDCG:

$$TDCG@x = \sum_{t=1}^x \frac{\gamma(y^x)}{\log(t+1)}, \quad (10)$$

where $x$ is the truncation position, $\gamma$ can add penalty items to low-relevant documents. High-relevant documents bring higher TDCG, while low-relevant documents will reduce TDCG. This reward mechanism enables the model to learn the optimal truncation point so that the returned list contains as few low-relevant documents as possible while retaining high-relevant documents. For the $T$-$th$ step, given reranked list $D' = \{r_1^1, r_1^2, ..., r_1^{T-1}\}$ from 1 to $T-1$, output document $r_1^T$, and the list of documents $R' = \{r_2^T, r_3^T, ..., r_{\beta+1}^T\}$ obtained from the local backward window, the local reranked list can be defined as $\{r_1^1, r_1^2, ..., r_1^{T-1}, r_1^T, r_2^T, ..., r_{\beta+1}^T\}$. If the model truncates the reranked list at the current step ($r_1^T$), the reward is TDCG@$T$, otherwise, the reward is TDCG@$(T+\beta)$, $\beta$ is the size of local backward window. Binary soft label at step $T$ is:

$$y_{cut}^T = \frac{\exp(TDCG@T)}{\exp(TDCG@T) + \exp(TDCG@(T+\beta))},$$

$$y_{nocut}^T = \frac{\exp(TDCG@(T+\beta))}{\exp(TDCG@T) + \exp(TDCG@(T+\beta))}.$$

The loss function of truncation can be defined as:

$$\mathcal{L}_T = -\sum_{t=1}^N (y_{cut}^t \log(p_1^t) + y_{nocut}^t \log(p_0^t)),$$

where $p_1^t$ and $p_0^t$ are defined as Equ.(7).

In the first epoch of training, the model only learns to rerank. In this way, the model can obtain the basic ability to judge the

relevance of documents. In later epochs, the model learns to rerank and truncate alternately in batches. When the model learns to rerank, the parameters of truncation module are fixed and $\mathcal{L}_R$ is the objective. When the model learns to truncate, the parameters of cross ranking FFN are fixed and $\mathcal{L}_T$ is the objective.

In inference, the reranked list is generated step by step and the ordinal number of the step is the rank of the document in final list. At each generation step, the model selects the best output document at current position based on the global and sequential dependency and makes the truncation decision concurrently. The final generated sequence is the reranked and truncated list with decreasing relevance. GenRT can be applied to scenarios that only require reranking or truncation. When the IR system does not need truncation, the truncation result can be not considered directly. When the IR system does not need reranking, the scoring matrix S at each step can be obtained from the input ranked list. For the scenarios that only require reranking, we propose an acceleration strategy that balances latency and accuracy. In training, it follows the generation paradigm step by step as described above. In inference, it directly uses the trainable vector *start* as the sequential dependency and uses the scoring matrix $S$ at the first step as the reranking results without generating sequence.

## 4 EXPERIMENTS

### 4.1 Experiment Settings

***Datasets.*** Datasets in our experiments can be divided into two categories: (1) Learning-to-rank public benchmarks for web search including Microsoft LETOR 30K (MSLR30K) [34], Yahoo! LETOR set 1(Yahoo!)[2] and Istella LETOR (Istella)[3]. These three datasets are collected from real search engines and they are feature-based. Each sample in these datasets is a feature vector, and the label has five-level relevance annotation from 0 (irrelevant) to 4 (perfectly relevant). (2) Open-domain Question-Answering datasets including Natural Queations [25] and TriviaQA [21]. These two datasets are text-based and the label has 2-level relevance annotation from 0 (irrelevant) to 1 (relevant). They are used to measure the reranking and truncation models on retrieval-augmented LLMs.

***Baslines and Evaluation Metrics.*** We select the SOTA models for reranking and truncation respectively as the baselines. For reranking, we select the following SOTA list-aware reranking models: **GlobalRerank** [49], **DLCM** [1], **Seq2Slate** [5], **GSF** [2], **PRM** [32], **SetRank** [31], **CRUM** [44], **SRGA** [33]. DASALC [35] is not compared because it is a much larger model (50 times the number of parameters of GenRT). Methods based on gradient boosting tree are not considered because they can not be applied to text-based data. For truncation, **BiCut** [26], **Choppy** [4], **AttnCut** [43] and **LeCut** [29] are selected as the baselines. We also introduce **Fixed**-$x$ that truncates the given ranking list at the fixed position $x$. Some retrieval-augmented works such as FID, REALM and Retro [7, 16, 19] are not considered. Because they need to train language models. Our work only focuses on the training of the IR models and does not require the training of language models, which can be flexibly compatible with black-box LLMs.

[2]http://learningtorankchallenge.yahoo.com
[3]http://blog.istella.it/istella-learning-to-rank-dataset/

For the metrics on three web search datasets, Normalized Discounted Cumulative Gain (NDCG) [20], Expected Reciprocal Rank (ERR) [8] and Mean Average Precision (MAP) are used to evaluate the performance of reranking [1, 31]. Following [43], TDCG (Equ.(10)) are used to evaluate the performance of truncation. $\gamma$ in TDCG (Equ. (10)) means that if the label is 0, outputs -4, if the label is 1, outputs -2, otherwise (2, 3, 4) outputs the label itself.

For the metrics on retrieval-augmented LLMs, improving the performance of LLMs is the ultimate goal of the retrieval system, so we use the accuracy of LLMs in answering open-domain questions as the evaluation metric. Since the labels in open-domain QA datasets are binary categories, $\gamma$ in TDCG (Equ. (10)) is set that if the label is 0, outputs -1, if the label is 1, outputs 1.

***Implementation.*** MHSA for global dependency encoder has 2 blocks and for sequential dependency decoder has 1 block. Each block has 8 heads and 256 hidden units. Compared with SetRank (6 blocks), our method has fewer parameters, which alleviates the inference overhead caused by generative paradigm to some extent. The shapes of transfer layer, $FFN$ and $rFFN$ are $[Z, 256]$, $[Z, 256]$ and $[512, 32, 1]$ respectively where $Z$ is the dimension of the input feature vector. The interaction-based ranker model for retrieval-augmented LLMs is *bert-base* [12]. The $\beta$ of local backward window is set as 4. The hyperparameter $\eta$ used to balance the two reranking loss is set as 0.1. In training, the batch size is 16 and Adam [24] with learning rate $10^{-5}$ is used to optimize the loss. We implement our model in PyTorch. Our method has the following implementations: **GenRT**: Jointly trained Reranking-Truncation model. **GenRT**$_{fast}$: Same training method as **GenRT** but uses the acceleration strategy that directly uses the scoring matrix $S$ at the first step as the reranking results in inference. **GenRT- w/o T**: Only learns to rerank. **GenRT- w/o R**: Only learns to truncate.

### 4.2 Performance on Web Search

This section evaluates the performance of GenRT and baselines on three learning-to-rank benchmarks collected from web search engine. Specifically, we follow the settings in SetRank [31] and DLCM [1] that use LambdaMart implemented by RankLib to retrieve the top 40 documents for each query as the input ranked lists. The lists are used as the input for list-aware reranking models.

***List-aware Reranking Performance.*** Table 1 shows that compared with the baselines, GenRT achieves the best list-aware reranking performance on three IR benchmark datasets. In the training, GenRT jointly learns to rerank and truncate end-to-end, in the inference, we decouple the two tasks (i.e., do not truncate the list) and use NDCG to evaluate its reranking performance. GenRT is better than GenRT-w/o T shows that joint modeling of reranking and truncation facilitates the sharing of contextual information and improves the performance of reranking. GenRT-w/o T outperforming most baselines (except SRGA on Yahoo!) indicates the effectiveness of integrating global and sequential dependency to represent the list-level contextual features and the sequence generation paradigm. GenRT$_{fast}$ is an acceleration strategy and outperforms most baselines with the faster inference than SetRank, which indicates that GenRT is a highly flexible model that achieves improvements for efficiency and performance in single reranking scenario.

**Table 1: Performance of different reranking models on three learning-to-rank benchemark datasets (bold: best; underline: runner-up; †: results with significant performance improvement with p-value ≤ 0.05 in T-test compared with baselines).**

| Reranker | MSLR 30K | | | | | Yahoo! | | | | | Istella | | | | |
|---|---|---|---|---|---|---|---|---|---|---|---|---|---|---|---|
| | NDCG | | ERR | | MAP | NDCG | | ERR | | MAP | NDCG | | ERR | | MAP |
| | @5 | @10 | @5 | @10 | - | @5 | @10 | @5 | @10 | - | @5 | @10 | @5 | @10 | - |
| GlobalRerank | 0.4501 | 0.4689 | 0.3445 | 0.3617 | 0.5110 | 0.6983 | 0.7425 | 0.4375 | 0.4514 | 0.7101 | 0.6190 | 0.6679 | 0.7001 | 0.7101 | 0.6840 |
| DLCM | 0.4500 | 0.4690 | 0.3440 | 0.3620 | 0.5109 | 0.6990 | 0.7430 | 0.4380 | 0.4517 | 0.7105 | 0.6194 | 0.6680 | 0.7005 | 0.7104 | 0.6843 |
| Seq2Slate | 0.4533 | 0.4701 | 0.3473 | 0.3685 | 0.5170 | 0.6993 | 0.7438 | 0.4385 | 0.4523 | 0.7143 | 0.6201 | 0.6693 | 0.7012 | 0.7114 | 0.6851 |
| GSF | 0.4151 | 0.4374 | 0.3215 | 0.3479 | 0.5073 | 0.6838 | 0.7316 | 0.4273 | 0.4405 | 0.7092 | 0.5968 | 0.6508 | 0.6822 | 0.7015 | 0.6805 |
| PRM | 0.4435 | 0.4602 | 0.3402 | 0.3550 | 0.5112 | 0.7072 | 0.7500 | 0.4390 | 0.4528 | 0.7147 | 0.6189 | 0.6605 | 0.6901 | 0.7080 | 0.6842 |
| SetRank | 0.4515 | 0.4696 | 0.3458 | 0.3632 | 0.5143 | 0.7029 | 0.7453 | 0.4380 | 0.4525 | 0.7140 | 0.6345 | 0.6834 | 0.7103 | 0.7273 | 0.6995 |
| CRUM | 0.4603 | 0.4812 | 0.3523 | 0.3745 | 0.5171 | 0.7078 | 0.7486 | 0.4397 | 0.4532 | 0.7150 | 0.6417 | 0.6902 | 0.7245 | 0.7401 | 0.7084 |
| SRGA | 0.4449 | 0.4672 | 0.3420 | 0.3619 | 0.5120 | 0.7079 | 0.7502 | 0.4400 | 0.4541 | 0.7159 | 0.6235 | 0.6713 | 0.7039 | 0.7120 | 0.6877 |
| GenRT | **0.4757**† | **0.4919**† | **0.3623**† | **0.3805**† | **0.5200**† | **0.7085**† | **0.7505**† | **0.4408**† | **0.4550**† | **0.7172**† | **0.6535**† | **0.7032**† | **0.7418**† | **0.7479**† | **0.7329**† |
| GenRT- w/o T | 0.4662 | 0.4878 | 0.3507 | 0.3760 | 0.5170 | 0.7068 | 0.7492 | 0.4392 | 0.4539 | 0.7145 | 0.6520 | 0.7018 | 0.7398 | 0.7450 | 0.7253 |
| GenRT$_{fast}$ | 0.4698 | 0.4891 | 0.3542 | 0.3780 | 0.5179 | 0.7072 | 0.7498 | 0.4394 | 0.4540 | 0.7161 | 0.6529 | 0.7023 | 0.7405 | 0.7462 | 0.7275 |

**Table 2: Performance of different truncation models on three learning-to-rank benchemark datasets (bold: best; underline: runner-up; †: results with significant performance improvement with p-value ≤ 0.05 in T-test compared with baselines).**

| Reranker | Truncation | MSLR | Yahoo! | Istella |
|---|---|---|---|---|
| | | TDCG@x | | |
| | Fixed-$x$ ($x$=5) | -1.15 | 1.14 | 4.50 |
| | Fixed-$x$ ($x$=10) | -3.29 | 0.50 | 3.55 |
| | BiCut | 0.12 | 2.55 | 3.22 |
| | Choppy | 0.33 | 2.63 | 3.77 |
| GenRT | AttnCut | 0.42 | 2.89 | 4.40 |
| | LeCut | 0.43 | 2.91 | 4.42 |
| | LeCut+JOTR | 0.54 | 2.93 | 4.45 |
| | GenRT- w/o R | 0.61 | 3.01 | 4.48 |
| | GenRT (End-to-End) | **0.84**† | **3.11**† | **5.03**† |

**Table 3: Reranking on retrieval-augmented LLMs. The input lists for list-aware reranker are retrieved by Contriever and ranked by interaction-based ranker. Acc. is for LLM.**

| Reranker | NQ | | | TriviaQA | | |
|---|---|---|---|---|---|---|
| | R@5 | R@10 | Acc. (LLM) | R@5 | R@10 | Acc. (LLM) |
| - | 59.40 | 77.31 | 57.63 | 68.22 | 85.24 | 62.70 |
| Seq2Slate | 60.15 | 77.50 | 58.05 | 68.90 | 85.47 | 63.55 |
| SetRank | 60.02 | 77.48 | 57.92 | 68.74 | 85.32 | 63.29 |
| GenRT | **60.78**† | **77.63**† | **58.79** | **70.01**† | **85.70**† | **64.37** |

*List-aware Truncation Performance*. Table 2 shows the truncation performance of GenRT and previous SOTA models. As GenRT is the best reranker demonstrated in Table 1, we use its reranking results as the input for the previous baselines models to achieve the fair comparison. The experimental results indicate that Reranking-Truncation joint model GenRT gets the best truncation performance. Specifically, GenRT outperforms GenRT- w/o R demonstrates the positive effect of joint modeling on truncation. GenRT- w/o R working better than previous SOTA models indicates the positive effects of integrating global and sequential information to make fine-grained truncation decision step by step and using RAML-based local binary classification soft criterion as the objective functions. In addition, our method outperforming LeCut+JOTR (a method that jointly trains reranking and truncation but still treats them as separate models and stages) shows the effectiveness of integrating the two tasks into a joint model and performing them concurrently.

## 4.3 Performance on Retrieval-augmented LLMs

This section evaluates the performance of GenRT and baselines on open-doma QA datasets under the retrieval-augmented LLMs

settings. We use Wikipedia passage-collection provided by [22] as the corpus, use Contriever [18] and interaction-based ranker [9] to perform retrieval-ranking pipeline to get top 40 passages as the input ranked list for list-aware reranking models. We use *gpt-3.5-turbo-16k* as the LLM. We provide the returned passage list from IR system to LLM in prompt and let LLM answer questions. We use EM [37] to count the accuracy of LLM answering questions by referring to the returned passage list.

*List-aware Reranking Performance*. Table 3 shows that the list reranked by GenRT can help LLM achieve better performance than retrieval-ranking pipeline and other list-aware reranking baselines[4]. The reason is that our method better reranks the passages in list to make relevant passages appear more at the top of the ranked list (higher Recall@5 and Recall@10). It means relevant information appears more at the start of the text input to LLM. There has been study [27] proves that LLMs prefer to exploit information at the start of the input text for generation, which can support our conclusion.

*List-aware Truncation Performance*. Table 4 shows that our method beats all baselines in balancing the number of passages in the retrieved list and the performance of LLMs. Specifically, as GenRT is the best reranker demonstrated in Table 1 and 3, we use its reranking results as the input for all truncation models to achieve

---

[4]Since most of the list-aware reranking models in the baselines are designed for feature-based data, we only reproduce SetRank and Seq2Slate that are suitable for text-based data (open-domain QA) and perform well in Table 1.

**Table 4: Truncation on retrieval-augmented LLMs. Length is the number of passages in the list. Acc. is for LLM.**

| Truncation | NQ | | | TriviaQA | | |
|---|---|---|---|---|---|---|
| | TDCG ↑ | Length ↓ | Acc. ↑ | TDCG ↑ | Length ↓ | Acc. ↑ |
| Fixed-$x$ ($x$=5) | -0.78 | 5.00 | 54.80 | 0.23 | 5.00 | 60.03 |
| Fixed-$x$ ($x$=10) | -0.95 | 10.00 | 55.72 | -0.17 | 10.00 | 61.19 |
| Fixed-$x$ ($x$=20) | -1.67 | 20.00 | 56.98 | -1.10 | 20.00 | 62.35 |
| Fixed-$x$ ($x$=30) | -4.78 | 30.00 | 56.05 | -2.34 | 30.00 | 62.30 |
| Fixed-$x$ ($x$=40) | -5.05 | 40.00 | 58.20 | -3.46 | 40.00 | 63.17 |
| BiCut | -0.35 | 22.75 | 56.79 | 0.38 | 25.83 | 62.30 |
| Choppy | -0.20 | 25.43 | 57.01 | 0.40 | 29.72 | 62.42 |
| AttnCut | -0.21 | 17.70 | 56.95 | 0.42 | 21.96 | 62.40 |
| LeCut+JOTR | -0.15 | 20.21 | 57.84 | 0.55 | 22.50 | 62.89 |
| GenRT | **-0.06**$^{\dagger}$ | 17.25 | 58.15 | **0.74**$^{\dagger}$ | 22.19 | 63.25 |

the fair comparison. Compared with Fixed-40, our method achieves comparable accuracy with much shorter retrieved list length. Compared with Fixed-20, our method achieves better accuracy with comparable length. This shows that our method effectively dynamically determines the suitable length of the retrieved list for each query, achieving the balance between efficiency (longer input list will increase the computational overhead of LLM) and accuracy.

## 4.4 Analysis

***Effect of Generative Ranking***. We explore the effect of generative ranking paradigm. One of the significant differences between GenRT and other methods is that it adopts global dependency encoder and sequential dependency decoder to combine list-level contextual features to generate the final reranked list step by step in the paradigm of sequence generation. The intuition is that the high-relevance documents (easy to distinguish) that have been generated are used as sequential dependency to control the subsequent document selection, which makes it easier for the model to distinguish confusing candidate documents. Figure 5(a) confirms our intuition. The solid line shows minimum distance between positive and negative samples in GenRT (the result of the lowest-scoring positive sample minus the highest-scoring negative sample in the scoring matrix $S$ (obtained by Equ.(6))). The dashed line is the corresponding distance estimated by SetRank. The distance of GenRT is bigger than the distance of SetRank and increases with step $T$, which indicates that sequential dependency makes relevant documents easier to be selected.

***Size of Local Backward Window***. Figure 5(b) shows that $\beta$ (the size of local backward window) affects the truncation performance. The results of the three datasets show that when $\beta$ is from 0 to 4, performance gets better because more valuable backward information is introduced, when $\beta$ is greater than 4, the performance deteriorates as $\beta$ increases because enlarging the window introduces more noise information caused by local ranking, which is not conducive to fine-grained modeling the truncation information.

***Loss Function in Reranking***. In Section 3.3, we design two loss functions and combine them to optimize reranking. In this section, we do the ablation study on Istella for them and results in Table 5 show that $\mathcal{L}_{sa\text{-}att}$ plays the most critical effect, and $\mathcal{L}_{sbs}$ further improves the reranking performance on the basis of $\mathcal{L}_{sa\text{-}att}$.

**Table 5: Ablation study for two loss functions of reranking.**

| Reranker | NDCG@1 | NDCG@5 | NDCG@10 |
|---|---|---|---|
| GenRT | 0.6911 | 0.6535 | 0.7032 |
| - w/o $\mathcal{L}_{sa\text{-}att}$ | 0.6089 | 0.5748 | 0.6273 |
| - w/o $\mathcal{L}_{sbs}$ | 0.6807 | 0.6436 | 0.6954 |

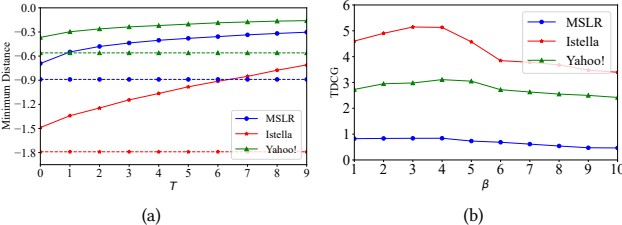

(a)                                              (b)

**Figure 5: (a) Minimum distance varies with generation step $T$ (solid line: GenRT, dashed line: SetRank). (b) TDCG varies with local backward window size.**

***Efficiency Analysis of List-aware Retrieval***. Compared with the reranking and truncation models that directly output the scores of all candidate documents (e.g. SetRank), GenRT generates the final list step by step. This leads to the increase in inference time of the IR system. To alleviate this problem, we propose two acceleration strategies. One is described in Section 3.3 that for the scenarios that only require reranking without truncation, the scoring matrix $S$ at the first step can be used as the reranking result directly and we call it GenRT$_{fast}$. The other is that we use fewer parameters than SetRank, especially for the decoder. The efficiency evaluation result performed on one Tesla V100 shows that for the inference time of reranking, GenRT$_{fast}$ is 0.6 times shorter than SetRank and GenRT is 2.1 times longer than SetRank. For the inference time of reranking-truncation pipeline, GenRT is 1.6 times longer than SetRank-AttnCut. Different from the efficiency requirements of first-stage retrieval, reranking and truncation have a much smaller candidate set and pay more attention to the performance of tasks. Considering the performance of reranking and truncation in Table 1, 2, 3 and 4, GenRT achieves significant performance improvement at the cost of a little longer inference time and has a positive effect on saving the computational overhead of LLM.

## 5 CONCLUSION

In this paper, we propose a Reranking-Truncation joint model called GenRT for list-aware retrieval in web search and retrieval-augmented LLMs. We adopt the structure of global dependency encoder and sequential dependency decoder to fully capture the list-level contextual features of the ranked list and share the information on reranking and truncation. We transform truncation into a step-by-step binary classification task and leverage the paradigm of sequence generation to combine dynamic reranking with static truncation. Experimental results on public learning-to-rank benchmarks and open-domain QA tasks show that our method achieves state-of-the-art performance on both reranking and truncation tasks for web search and retrieval-augmented LLMs.

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

# A   APPENDIX

## A.1   Details of Datasets

**Table 6: Details of feature-based datasets for web search.**

| Dataset | #Queries Train | #Queries Test | #Doc Train | #Doc Test | #Feature |
|---|---|---|---|---|---|
| MSLR30K | 18,919 | 6,306 | 2,270k | 753k | 136 |
| Yahoo! | 19,944 | 6,983 | 473k | 165k | 700 |
| Istella | 20,317 | 9,799 | 7,325k | 3,129k | 201 |

Table 6 shows the details of feature-based datasets for web search. Each sample in these datasets is a feature vector extracted by traditional learning-to-rank method such as matching, BM25 and pagerank.

## A.2   Relevance Capturing in Truncation

We explore the ability of truncation models to capture the relevance of documents. The relevance of the documents at the truncation point (i.e, the first truncated document) can reflect the ability of the truncation model to understand document relevance. On the one hand, if the document at the truncation point is high-relevant, it means that the truncation model misunderstands the relevance of

the document and truncates the high-relevant document, leading to the negative impact on the metric (i.e. DCG). On the other hand, if the document at the truncation point is low-relevant, it at least proves that the truncation model can distinguish low-relevant documents. Figure 6 shows the relevance distribution of the documents at the truncation point obtained from different models. The result shows that the documents at the truncation point obtained from GenRT have lower relevance, which indicates that GenRT has the strongest ability to capture the relevance of documents in truncation compared with the other baselines. This performance comes from joint modeling of reranking and truncation, which enables the truncation to take full advantage of the modeling information of document relevance in reranking. While separation between reranking and truncation makes the two cannot share information well.

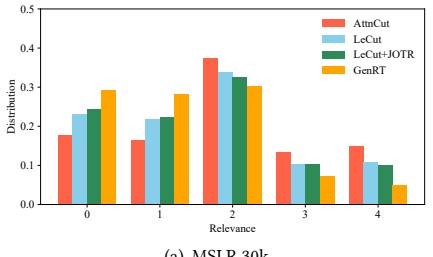
(a)  MSLR 30k

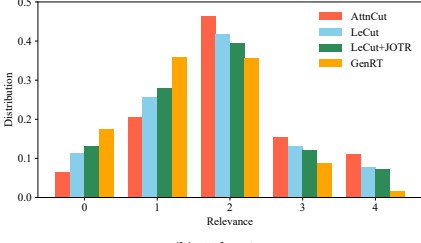
(b)  Yahoo!

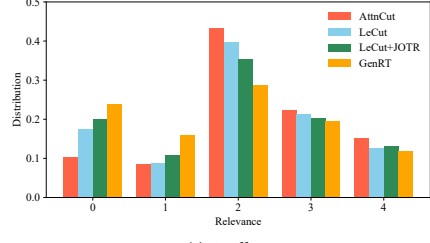
(c)  Istella

**Figure 6: Relevance distribution of the first truncated documents. X-axis is the relevance. Y-axis represents the distribution of the first truncated document with the corresponding relevance out of all first truncated documents. For low-relevant (0 and 1) documents, the higher the value the better, for high-relevant (2, 3 and 4) documents, the lower the value the better.**

