# OpenReview forum: "List-aware Reranking-Truncation Joint Model for Search and Retrieval-augmented Generation"
_ACM.org/TheWebConf/2024/Conference — TheWebConf24 Oral_

### Official Review · Reviewer_fzkQ · 2023-11-02

**Novelty:** 5
**Technical Quality:** 5

**Review:**

The paper proposes a joint model called GenRT for learning and performing reranking and truncation concurrently in list-aware retrieval systems. The key idea is using an encoder-decoder architecture where the encoder captures global list-level features and the decoder generates the reranked list step-by-step while making truncation decisions. The parameters of the encoder are shared in the joint optimization, and to learn the truncation model in the dynamic reranking process, a local backward window is introduced to provide backward context. A couple of reranking and truncation losses are designed to train the model for the two tasks. The authors have conducted extensive experiment with both LTR datasets and retrieval augmented LLMs on QA datasets.

Strengths:

- The paper is mostly easy to read.
- The authors have conducted extensive experiments to show the effectiveness of their methods
- Overall, the proposed method is reasonable and the experiment results support the authors’ arguments in the paper. The learning of truncation with dynamic list is particularly interesting.

Weakness:

- The idea of jointly training a reranking model with a truncation model is good but a bit incremental since LeCut has already considered jointly training the ranking models with the truncation models. The authors have clearly explained their differences with LeCut, but the differences are mostly on the model side, which means that main framework is not surprising new in this paper.
- The proposed TDCG reward/metric is not full grounded by user studies or previous studies. This could be problematic considering that it is the only truncation metric used in this paper. Particularly, there is no justification on how the gamma is selected.

**Questions:**

- Section 3.1, what is the ranking score l_i in feature-based datasets? The ranking score from the initial list? Or the position in the initial list?
- NCI is a widely used metric in truncation. Why not use NCI but proposed a new metric TDCG?
- Line 643, does \gamma means \gamma(y*x)?
- In Table 4, what does’t mean to compute TCDG for retrieval-augmented LLMs? How to compute it exactly?

**Reviewer Confidence:**

4: The reviewer is certain that the evaluation is correct and very familiar with the relevant literature

**Scope:**

4: The work is relevant to the Web and to the track, and is of broad interest to the community

---

### Official Review · Reviewer_da2Z · 2023-11-24

**Novelty:** 5
**Technical Quality:** 5

**Review:**

This paper proposes GenRT, which jointly does reranking and ranking truncation in one model structure. GenRT exploits global dependency encoder to capture the information of the whole ranked list and decide which document to pick and whether to be truncated simultaneously in each step of the sequential decoding phase.

Positive Feedback:

1. The proposed model, GenRT, effectively combines reranking and ranking truncation within a novel model structure.

2. The presentation of the paper is clear, with a well-structured description and helpful visualizations.

3. The inclusion of baselines for both reranking and truncation, along with comparisons in different settings (w/o T, w/o R), demonstrates the behavior and flexibility of the proposed model.

Concerns and Suggestions:

1. For experiment results, while statistical significance is indicated, it would be beneficial to elaborate on the actual impact of the observed improvements, particularly in cases where they are small, such as the reranking results for Yahoo! and reranking results on retrieval-augmented LLMs for NQ.

2. In the results of truncation, it would be valuable to include the TDCG of optimal truncation to provide a deeper understanding of the performance of truncation models. Additionally, including the truncation performance of GenRT with other reranking models can offer insights into GenRT's truncation performance without the benefit of the end-to-end process and the same model structure.

3. Given the proposed acceleration strategy, it is suggested to include an efficiency analysis in the experimental results to provide a comprehensive understanding of the model's efficiency.

4. According to Table 4, it is noted that the highest accuracy for NQ is achieved on the fixed-x setting with the largest x. To gain a more comprehensive understanding of truncation performance, it would be valuable to explore the performance of all models with larger x which causes worse accuracy. This analysis can provide insights into the value of truncation.

5. The symbol 'p' is used in equations 7 and 9 to represent different meanings.

**Questions:**

It would be helpful if the authors could address the concerns about model performance and evaluation.

**Reviewer Confidence:**

3: The reviewer is confident but not certain that the evaluation is correct

**Scope:**

4: The work is relevant to the Web and to the track, and is of broad interest to the community

---

### Official Review · Reviewer_F3xg · 2023-11-25

**Novelty:** 5
**Technical Quality:** 5

**Review:**

The study introduces list-aware retrieval, which involves presenting search results as a ranked list of documents. The tasks involved are reranking and truncation. Previous studies treated them separately, but the proposed joint model, GenRT, performs them concurrently using a generative paradigm. This approach improves information sharing and addresses the issue of error accumulation. Experimental results show that GenRT achieves better performance in both reranking and truncation tasks for web search and retrieval-augmented language models.

Pros:
1.The proposed combined training of document reranking and truncation appears to be novel compared to existing approaches.
2.The experimental design clearly demonstrates that the combination of reranking and truncation improved both tasks on the information retrieval datasets.
3.The paper does a decent job of describing their models and experiments. Readers should be able to reproduce their work based on the details provided in the paper.

Cons:
1.There are some parts of the writing that require clarification.
2.The improvement on the information retrieval dataset appears to be moderate.
3.The application of truncation to the large language model question-answering (LLM QA) task seems to reduce the algorithm's performance.

**Questions:**

1.It would be beneficial to define and explain truncation and retrieval-augmented LLMs earlier, such as in the introduction or abstract section.
2.Justification for selecting learning-to-rank and QA as testing tasks should be provided earlier in Section 1.
3.The meaning of "they do not satisfy the permutation invariant" in Section 2 is unclear.
4.Additional details on how document embeddings are obtained are necessary in Section 3.1.
5.The importance and necessity of position embedding should be clarified, and supporting experiments could be added.
6.Further justification is needed for using TDCG as an evaluation metric and how it penalizes irrelevant documents.
7.In Section 4.4, an explanation is needed for why SetRank is a suitable baseline in Fig 5.
8.Table 4 seems to suggest that applying truncation negatively impacts LLM performance on QA tasks. This makes people question the importance of the truncation task.

**Reviewer Confidence:**

3: The reviewer is confident but not certain that the evaluation is correct

**Scope:**

3: The work is somewhat relevant to the Web and to the track, and is of narrow interest to a sub-community

---

### Official Review · Reviewer_5vwi · 2023-11-27

**Novelty:** 5
**Technical Quality:** 5

**Review:**

Summary
----
The article introduces the GenRT model, a novel approach that simultaneously addresses reranking and truncation through a generative framework built on an encoder-decoder architecture.

Strong Points
----
1. The innovative integration of reranking with truncation tasks.
2. Demonstrated improvements in performance through tests on widely available datasets.
3. The paper is clear to read and follow.

Weak Points
----
1. A key area of concern with the GenRT model is its application in real-time or online environments, where retrieval-augmented generation is usually used. The paper does not adequately address the potential increase in computational complexity and latency that the model might introduce in such scenarios. This oversight is significant, as it leaves questions about the model's practicality and efficiency in real-world applications. I would suggest the authors to include a detailed comparison of the GenRT model's latency and complexity against existing baseline methods. Such an analysis would provide a clearer understanding of the model's performance in time-sensitive applications. This additional analysis could offer more comprehensive insights into the trade-offs between model performance and operational efficiency.

**Questions:**

See weak points.

**Reviewer Confidence:**

3: The reviewer is confident but not certain that the evaluation is correct

**Scope:**

4: The work is relevant to the Web and to the track, and is of broad interest to the community

---

### Official Review · Reviewer_ZWGp · 2023-11-29

**Novelty:** 5
**Technical Quality:** 6

**Review:**

This paper proposes a Reranking-Truncation joint model for list-aware retrieval in web search and retrieval-augmented LLM. The experimental results have verified the effectiveness of the method, and the experimental design is reasonable.
This paper has a well-organized structure.
Truncation and Reranking are important problems in retrieval and integrating two parts together is an important issue.
Efficiency Analysis of List-aware Retrieval in section 4.4 is very necessary.

**Questions:**

More detailed information is needed on the time-consuming data of truncation and reranking using traditional methods.

**Reviewer Confidence:**

3: The reviewer is confident but not certain that the evaluation is correct

**Scope:**

4: The work is relevant to the Web and to the track, and is of broad interest to the community

---

### Decision · Program_Chairs · 2024-01-22

**Decision:**

Accept (Oral)

**Comment:**

This is a metareview. It is based on the reviews, the author's feedback, and my own opinion. This paper proposes a list-aware reranking with early termination. While the reviews were reasonably detailed and pointed out important issues, it is my opinion that the authors did a very good job of carefully addressing all of these. I must admit that I am a little disappointed that several of the reviewers did not seem to acknowledge/engage with the the auth
 ors on their detailed responses, I want to commend the authors. It is my opi
 nion that this paper should be accepted.

 A few small comments regarding potential changes:
 * A few questions were raised regarding the efficiency / complexity aspect of the early termination mechanism.
  The authors provided a detailed response to this issue, which should be included in the camera ready.
 * Reviewer 3 has made a few concrete suggestions about the order of
  presentation that the authors should consider in the camera ready.
 * Other detailed experimental responses should be included in the camera
  ready if possible as they clearly improve the quality of the work in my
  opinion.

 This is a nice piece of work. Well done.